# Classification of Mine Drainages in Japan Based on Water Quality: Consideration for Constructed Wetland Treatments

Satoshi Soda [1],* and Thuong Thi Nguyen [2]

1 Department of Civil and Environmental Engineering, Ritsumeikan University, 1-1-1 Nojihigashi, Kusatsu 525-8577, Shiga, Japan

2 Asia-Japan Research Institute, Ritsumeikan University, 1-1-1 Nojihigashi, Kusatsu 525-8577, Shiga, Japan

* Correspondence: soda@fc.ritsumei.ac.jp

**Abstract:** As a passive technology, constructed wetlands (CWs) are promising candidates for mine-drainage treatment. However, the design and operation of CWs have not been fully established because the chemical compositions of mine drainage are diverse. In this study data sets of 100 mine drainages in Japan were classified using multivariate analysis based on water quality. Mine drainage was classified into eight types based on the ratio of the concentrations of Cd, Pb, As, Cu, Zn, Fe, and Mn to the effluent standard: (I) neutral and low metal concentration, (II) weakly acidic and low metal concentration, (III) weakly acidic and high Zn concentration, (IV) weakly acidic and high Mn and Zn concentrations, (V) acidic and high As concentration, (VI) acidic and high Fe concentration, (VII) acidic and extremely high Fe concentration, and (VIII) acidic and high Zn concentration. Mechanisms for removing metals in CWs were discussed based on this classification. Metal hydroxides of Cu, Pb, Zn, and Cd can precipitate with an increasing pH. Under oxidative conditions, dissolved Fe and Mn are oxidized to metal oxides. Under reductive conditions, Pb, Zn, Cd, and Cu precipitate as metal sulfides. This classification of mine drainage will be helpful in the systematic design and operation of CWs.

**Keywords:** principal component analysis; cluster analysis; effluent standard; hydroxides; sulfides





## 1. Introduction

When ores are exposed to rain and air, metals and metalloids such as iron (Fe), copper (Cu), lead (Pb), manganese (Mn), cadmium (Cd), zinc (Zn), and arsenic (As) are released into water. Mine drainage, which contains toxic metals, threatens ecosystems and human health. All major global mining countries have serious problems caused by acid mine drainage (AMD) with a pH value of less than 6.5 [1,2]. To prevent pollution, it is necessary to minimize mine-drainage generation. Thus, mine drainage should be treated conventionally with physicochemical processes such as neutralization, coagulation, sedimentation, and filtration, with high costs for energy and reagents [1,2].

In Japan, since the 1970s, approximately 5000 mines have been abandoned or closed because of increasing labor costs and the import liberalization of metal resources. Even after the mining operation is stopped, mine drainage containing toxic metals is practically permanent. Among these, approximately 80 legacy mines must continue mine drainage treatment [3,4]. Various strategies have been applied to reduce mine-drainage generation, including reforestation, tunnel plugging, and sludge backfill [4]. Furthermore, reliable and inexpensive processes are required for treating the generated mine drainage.

The Ministry of Economy, Trade, and Industry, Japan (METI) and the Japan Oil, Gas and Metals National Corporation (JOGMEC) published a guidance of passive treatment for mine-drainage treatment in 2021 [5,6]. As a passive technology, constructed wetlands (CWs) incorporating physical, chemical, microbial, and botanical processes are promising candidates for the sustainable treatment of mine drainage. CWs for AMD have been

reviewed by several researchers [7–10]. Figure 1 displays the conceptual mechanisms of metal removal in CWs. In CWs, the predominant mechanisms for metal removal are different under various types of mine drainage. With an increase in the pH, the metal hydroxides of Cu, Pb, Zn, and Cd precipitate in CWs filled with alkaline substrates such as limestone and seashells. Under oxidative conditions, dissolved Fe and Mn are biochemically oxidized to metal oxides, resulting in co-precipitation with other metals such as Zn and As. Under reductive conditions, Pb, Zn, Cd, and Cu precipitate as metal sulfides by the microbial mediation. The selection of wetland plants is eventually affected by the characteristics of the mine drainage. CWs are beneficial irrespective of the usage of other processes. As a pretreatment for physicochemical processes, CWs can adjust the pH value and reduce the drainage amount through evapotranspiration. As a post-treatment with a low operational cost, a CW can remove the remaining metals from the effluent of the major physicochemical processes.

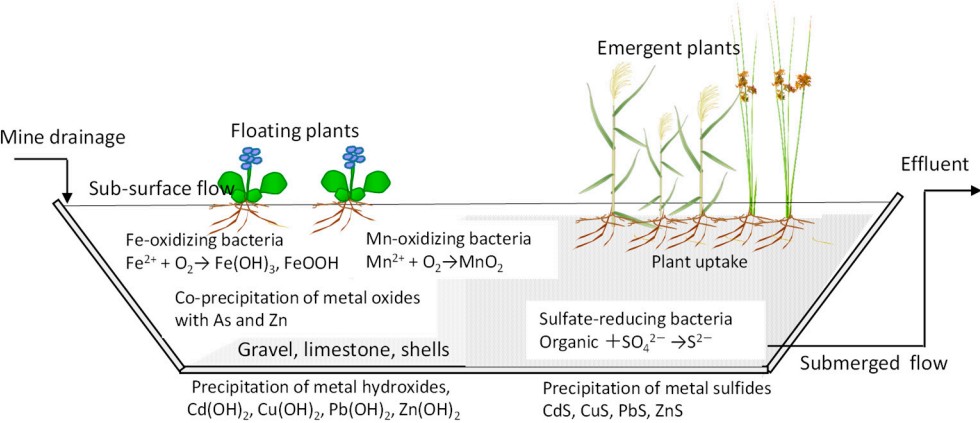

**Figure 1.** Conceptional mechanisms for metal removal in constructed wetlands.

However, only a limited number of CWs are used for mine drainage treatment in Japan. At Motokura Mine in the Hokkaido Prefecture, a CW was installed on a small scale in 2006 for removing Zn, Pb, and As [11,12]. A CW at Ningyotoge Mine in the Chugoku Region is also used for As removal from AMD through ferrihydrite coprecipitation [13]. The design and operation of CWs have not been completely established in Japan because the chemical composition of mine drainage is diverse. The Gray Acid Mine Drainage Index (AMDI) uses pH, sulfate, Fe, Zn, Al, Cu and Cd [14]. The framework by Hill [15] classifies AMD using acidity, sulfate, pH, Al, and Fe concentrations [15,16]. According to JOGMEC [17], although the quantitative backgrounds are unclear, mine drainage in Japan is generally classified into five types: (A) the acidic with Fe and base metals (Cu, Pb, Zn) predominant type, (B) the strongly acidic with Fe and As predominant type found in limonite mines, (C) the weakly acidic with Mn and base metals predominant type as observed in few Pb, Zn, and Mn mines; (D) the neutral with As predominant type observed in few gold and As mines, and (E) the neutral with Cd and Zn predominant type observed in few Pb, Zn, tin, and tungsten mines.

In this study, datasets of 100 mine drainages in Japan were quantitatively classified by multivariate analysis based on water quality. The purpose of this study was to propose a new classification of mine drainage for discussing the design and operation of CWs for removing heavy metals. The mine-drainage classification proposed in this study was compared with the conventional JOGMEC's classification. The mechanism for removing metals in CWs was discussed based on the mine-drainage classification.

## 2. Analytical Methods

The annual average water quality data for 2014, 2015, and 2016 (flow rate, pH, and concentrations of Cd, Pb, As, Cu, Zn, Fe, and Mn) from 100 mine drainages in Japan were

provided by METI. Each mine drainage had an ID number (1–100): 20 in Hokkaido, 39 in Tohoku, 6 in Kanto, 4 in Chubu, 9 in Kinki, 7 in Chugoku, 2 in Sikoku, and 13 in Kyushu [2]. The names and locations of the mines corresponding to individual drainages are concealed in the present study. The locations of 80 mines in Japan are shown elsewhere [3,4]. Hokkaido and Tohoku belong to the temperate zone with a cool, humid continental climate. The other regions belong to the temperate zone with a humid, subtropical climate. Many mines are located in mountains where the climate and weather are influenced by altitude.

The metal concentrations of each mine drainage over three years were averaged. The average concentrations were divided according to the effluent standards in Japan for standardization. The effluent standard is as follows: Cd: 0.03 mg/L, Pb: 0.1 mg/L, As: 0.1 mg/L, Cu: 3 mg/L, Zn: 2 mg/L, soluble Fe: 10 mg/L, soluble Mn: 10 mg/L. Standardized concentrations were used for principal component analysis (PCA) and cluster analysis using PAST ver. 1.3.4 software [18]. The pH values of the dataset were not used for the multivariate analysis, but the effluent standard for pH is 5.6–8.6 for discharging to lakes and rivers.

## 3. Results and Discussion

### 3.1. Characteristics of Mine Drainage

Figure 2 shows the distribution of flow, pH, and the ratio of the metal concentration to the effluent standard for 100 mine drainages. The flow rate was distributed in a wide range of $5$–$2.6 \times 10^4$ m$^3$/d (Figure 2A). Out of 100 mine drainages, the flow rates of 21 and 48 mine drainages were less than 150 m$^3$/d and 500 m$^3$/d, respectively. The hydraulic retention times of CWs utilized for mine drainage treatment typically range from a few days to several weeks. Considering the flow rates and retention times, the areas and volumes of CWs should be determined. Most CWs in the U.S. for mine-drainage treatment have been designed for a low flow rate of less than 150 m$^3$/d, with a maximum of 570 m$^3$/d [19].

Out of the 100 mine drainages, 48 displayed low pH values of less than 4.0, whereas 25 displayed neutral pH values of 5.6–8.6. No mine drainage demonstrated a pH value higher than 8.6 (Figure 2B). Low pH values suggest the presence of sulfate in mine drainage, although the dataset used in this study did not include its concentration. Pre-conditioning by the neutralization process is required for the application of a CW treatment to AMD. Otherwise, limestone, as an alkali, is implemented as a filter medium or substrate in CWs.

Out of the 100 mine drainages, 32 for Cd, 29 for Pb, 15 for As, 21 for Cu, 54 for Zn, 49 for Fe, and 13 for Mn showed concentrations higher than the effluent standard. A major treatment process is suitable for moderately polluted mine drainages that are one–ten times higher than the effluent standard: twenty-nine for Cd, twenty-six for Pb, six for As, eighteen for Cu, forty-two for Zn, twenty-eight for Fe, and thirteen for Mn (Figure 2C). For highly polluted mine drainage, CWs, as a post-treatment process, can remove the remaining metals from the effluent of the major physicochemical process.

### 3.2. Classification of Mine Drainage

### 3.2.1. Clusters I–VIII

Figure 3 displays the classification of mine drainage using PCA and cluster analysis based on standardized metal concentrations. Mine drainages No. 46 and No. 79 with high Fe and Cu concentrations demonstrated extremely high scores for the first principal component (PC1). The second principal component (PC2) was positive and negative for mine drainages with high Zn concentrations and high As concentrations, respectively. Mine drainage No. 9 displayed an extremely high score, whereas No. 3 and No. 94 displayed low PC2 scores.

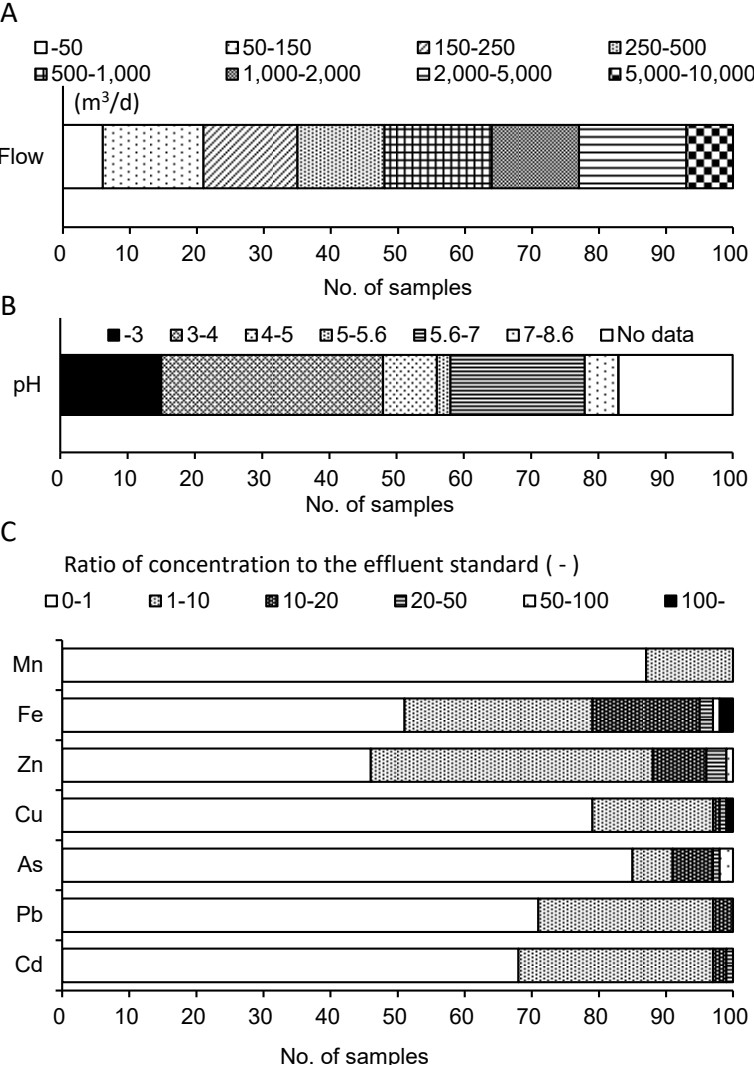

**Figure 2.** Average characteristics of 100 mine drainages in Japan (2014–2016). Flow rate (**A**), pH (**B**), and ratio of metal concentration to the effluent standard in Japan (Cd: 0.03 mg/L, Pb: 0.1 mg/L, As: 0.1 mg/L, Cu: 3 mg/L, Zn: 2 mg/L, soluble Fe: 10 mg/L, and soluble Mn: 10 mg/L) (**C**).

Except for such distinctiveness, mine drainages were arbitrarily divided into eight clusters, designated as Clusters I–VIII by constraint-based clustering based on the standardized metal concentrations. The characteristics of the clusters are shown in Figure 4 and Table 1. Cluster I (14 mine drainages) was neutral (weakly acidic–weakly alkaline) and demonstrated a low-metal-concentration type. Cluster II (24 mine drainages) was weakly acidic and demonstrated a low metal concentration. Cluster III (16 mine drainages) was weakly acidic and demonstrated a high Zn concentration. Cluster IV (four mine drainages) was weakly acidic and had high Mn and Zn concentrations. Cluster V (three mine drainages) was acidic and possessed a high As concentration. Cluster VI (nine mine drainages) was acidic and possessed a high Fe concentration. Cluster VII (six mine drainages) was acidic and possessed an extremely high Fe concentration. Cluster VIII (seven mine drainages) was acidic and displayed a high Zn concentration. Mine drainages No. 28 and No. 81 belonged to an upper cluster of Clusters I and II that commonly displayed 2.4 times higher Cu concentrations than the effluent standard with low pH values below three (Figures 3 and 4). Mine drainages No. 26 and No. 48 belonged to an upper cluster of Clusters I–IV that commonly demonstrated 2.8 and 8.5 times higher concentrations of Zn and Pb than those of the effluent standards, respectively.

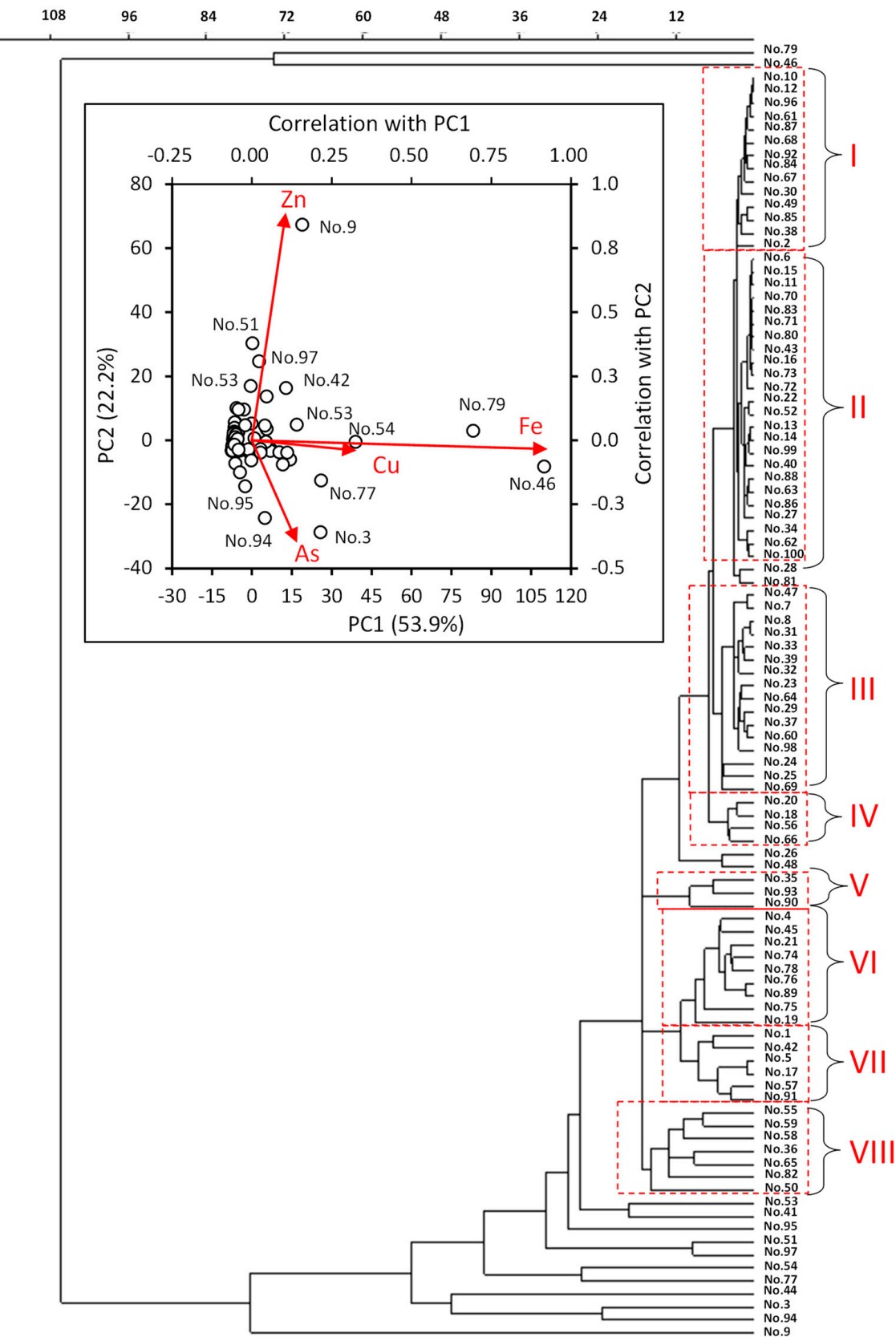

**Figure 3.** Classification of 100 mine drainages (No. 1–100) in Japan (2014–2016). Constraint-based clustering analysis classified mine drainages to clusters I–VIII and others. The inner panel shows PC scores 1 and 2 by PCA analysis of mine drainages.

**Table 1.** Classification of mine drainages in Japan (2014–2016) based on the cluster analysis (Figures 3 and 4).

| Cluster | Type | Mine Drainages | Cd, mg/L | Pb, mg/L | As, mg/L | Cu, mg/L | Zn, mg/L | Fe, mg/L | Mn, mg/L | pH | Note |
|---|---|---|---|---|---|---|---|---|---|---|---|
| I | Neutral (weakly acidic–weakly alkaline) and low metal concentration type | 14 (No. 2, 10, 12, 30, 38, 49, 61, 67, 68, 84, 85, 87, 92, 96) | 0.00–0.04 | 0.00–0.24 | 0.00–0.082 | 0.00–4.00 | 0.00–4.20 | 0.00–34.93 | 0.00–0.97 | 5.8–8.0 | Partly JOGMEC type E (Neutral and Cd and Zn predominant type) |
| II | Weakly acidic and low metal concentration type | 24 (No. 6, 11, 13, 14, 15, 16, 22, 27, 34, 40, 43, 52, 62, 63, 70, 71, 72, 73, 80, 83, 86, 88, 99, 100) | 0.00–0.26 | 0.00–0.13 | 0.00–0.04 | 0.00–1.67 | 0.00–4.30 | 0.00–28.32 | 0.00–7.97 | 3.0–5.3 | |
| III | Weakly acidic and high Zn concentration type | 16 (No. 7, 8, 23, 24, 25, 29, 31, 32, 33, 37, 39, 47, 60, 64, 69, 98) | 0.00–0.14 | 0.00–0.21 | 0.00–0.05 | 0.00–6.01 | 7.42–19.67 | 0.00–41.41 | 0.00–8.83 | 2.9–6.7 | |
| IV | Weakly acidic and high Mn and Zn concentration type | 4 (No. 18, 20, 56, 66) | 0.00–0.03 | 0.00–0.10 | 0.00–0.03 | 0.02–1.53 | 2.57–11.46 | 0.00–11.80 | 34.54–75.93 | 4.6–6.3 | JOGMEC type C (Weakly acidic and Mn and base metals predominant type) |
| V | Acidic and high As concentration type | 3 (No. 35, 93, 90) | 0.00–0.01 | 0.00–0.32 | 0.98–1.87 | 0.00–0.84 | 0.00–2.98 | 0.00–59.00 | 0.00–0.00 | 3.1–7.4 | Partly JOGMEC types B (Strongly acidic and Fe and As predominant type) and D (Neutral and As predominant type) |
| VI | Acidic and high Fe concentration type | 9 (No. 4, 19, 21, 45, 74, 75, 76, 78, 89) | 0.01–0.04 | 0.13–0.68 | 0.08–0.34 | 4.55–18.10 | 2.38–7.97 | 88.74–120.50 | 6.06–45.50, | 2.4–5.1 | JOGMEC type A (Acidic and Fe and base metals predominant type) |
| VII | Acidic and extremely high Fe concentration type | 6 (1, 5, 17, 42, 57, 91) | 0.00–0.01 | 0.08–0.26 | 0.27–0.92 | 3.54–10.73 | 0.98–3.32 | 185.94–214.33 | 3.55–10.81 | 2.3–3.9 | Partly JOGMEC type B |
| VIII | Acidic and high Zn concentration type | 7 (No. 36, 50, 55, 58, 59, 65, 82) | 0.09–0.19 | 0.54–1.28 | 0.01–0.07, | 3.87–9.37 | 28.14–43.63 | 52.69–118.03 | 13.63–56.26 | 2.7–5.1 | Partly JOGMEC type A |

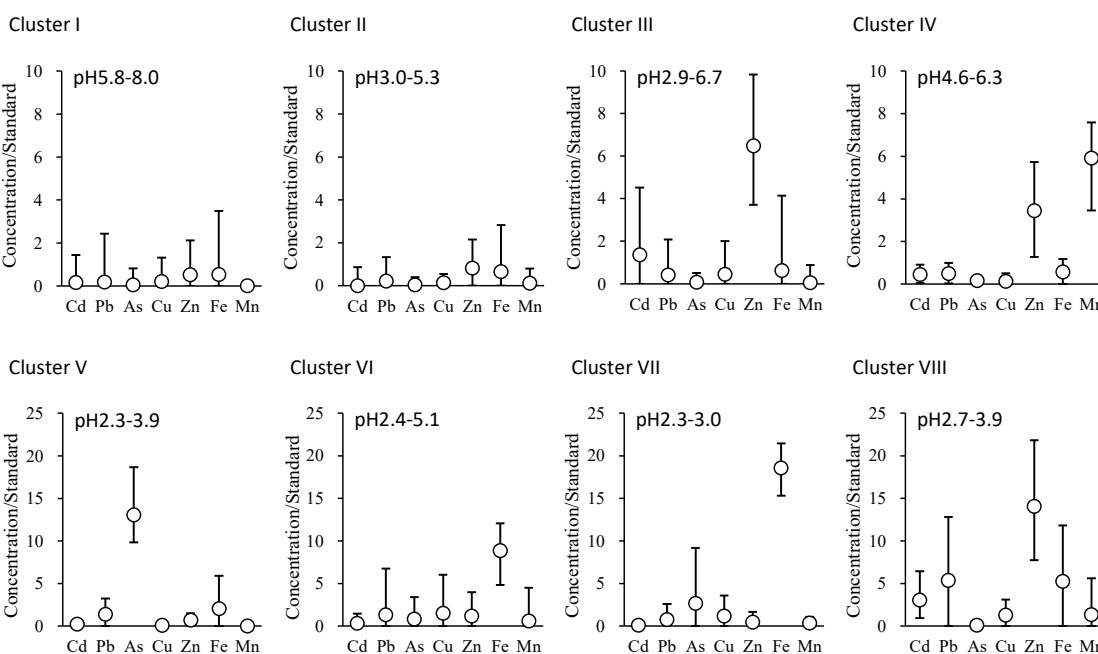

**Figure 4.** The ratio of the metal concentration to the effluent standard (Cd 0.03 mg/L, Pb 0.1 mg/L, As 0.1 mg/L, Cu 3 mg/L, Zn 2 mg/L, soluble Fe 10 mg/L, and soluble Mn 10 mg/L) in Clusters I–VIII of mine drainage in Japan (2014–2016) (Figure 3). Average plots and bars for maximum and minimum values are displayed in detail.

### 3.2.2. Comparison of Clusters I–VIII with JOGMEC Types

Clusters I–VIII were compared with the conventional JOGMEC types in Table 1. The distinctive mine drainages Nos. 9, 46, 53, 54, and 79 apparently correspond to JOGMEC type A. Mine drainages in Cluster VI and partly in Cluster VIII also correspond to JOGMEC type A. On the other hand, Nos. 3, 77, 94, and 95 correspond to JOGMEC type B. A few mine drainages in Clusters V and VII also correspond to JOGMEC type B. These highly polluted mine drainages are typically treated by neutralization using slaked lime and a high-density sludge-recycling method [20]. Although a few mine drainages exist with high Mn concentrations in Clusters VI and VIII, Cluster IV apparently corresponds to JOGMEC type C. Mine drainage No. 35 in Cluster V with a neutral pH value corresponds to JOGMEC type D. A few mine drainages of Cluster I correspond to JOGMEC type E. Interestingly, many mine drainages in Clusters II and III did not correspond to any JOGMEC type.

### 3.3. Consideration of CW Treatment for Classified Mine Drainages

#### 3.3.1. Cluster I

Mine drainages classified as Cluster I demonstrate neutral (weakly acidic–weakly alkaline) pH values, with one or two types of metals slightly exceeding the effluent standard. The weakly acidic mine drainage of Cluster I is typically treated by a pH increase with sodium hydroxide to precipitate metal hydroxides, such as $Cd(OH)_2$, $Cu(OH)_2$, $Pb(OH)_2$, and $Zn(OH)_2$, followed by neutralization using sulfate. The treatment processes for these mine drainages could be replaced with CWs on a priority basis. In fact, mine drainage in Cluster I in the Hokkaido Prefecture, a subsurface-flow CW planted with common reed (*Phragmites australis*), was installed on a small scale in 2006 to remove Zn, Pb, and As, and was scaled up for a demonstration test to replace the existing neutralization process [12].

#### 3.3.2. Cluster II

Mine drainages classified as Cluster II are weakly acidic and contain one or two types of metals slightly exceeding the effluent standard. The installation of a CW treatment should be considered with priority to mine drainage in Cluster II in addition to Cluster I. Presently,

these are mostly treated by neutralization using slaked lime or sodium hydroxide. For example, mine drainage No. 68 displayed low concentrations with other metals but slightly higher Cd concentrations than the effluent standard. A laboratory-scale experiment was conducted to evaluate the treatment performance of CWs filled with loamy soil for synthetic mine drainage, simulating the chemical composition [21]. The CWs removed Cd to satisfy the effluent standards from mine drainage, mostly by soil adsorption. The presence of emergent plants, cattail (*Typha latifolia* L.) and common reed enhanced metal removal by filtration with elongated roots. Furthermore, it enhanced metal sulfide precipitation by sulfate-reducing bacteria (SRB) in the rhizosphere. SRB decomposes organic matter into lower-molecular-weight acids and bicarbonate, leading to increased alkalinity and the formation of metal sulfide precipitates [22]. Bacteria–plant interactions are regarded as important because they imply symbiotic mechanisms for heavy metal removal and tolerance. Based on the lab-scale results, small pilot–scale CWs with a hydraulic retention time of 3.8–1.2 days were installed in the treatment plant for mine drainage No. 68. The unplanted and the cattail-planted CWs reduced the average concentrations of Cd from 0.031 to 0.01 and 0.005 mg/L [23].

### 3.3.3. Cluster III

Mine drainages classified as Cluster III contain high Zn and other metals with a low pH. For example, mine drainage No. 69 displayed a low pH of approximately four and two- and three-times higher concentrations of Zn and Fe, respectively, than the effluent standard values. A lab-scale batch experiment was conducted to evaluate the treatment performance of CWs filled with limestone and oyster shells and planted with cattails for actual and synthetic mine drainage [24]. Seashells as byproducts of aquaculture are composed mainly of non-hazardous calcium carbonate. Moreover, Zn can be removed by the precipitation of hydroxide $Zn(OH)_2$ at a pH higher than 7.8. Anaerobic bioreactors are also effective at removing Zn at such high concentrations. A laboratory-scale SRB bioreactor containing rice bran and husk was successfully operated to treat AMD containing 15 mg/L Zn, 40 mg/L Fe, 5 mg/L Cu, and 0.06 mg/L Cd [22].

### 3.3.4. Cluster IV

Mine drainages classified as Cluster IV indicated 3–7-times higher Mn concentrations than that of the effluent standard, and 1.5–6-times higher Zn concentration than that of the effluent standard. Using synthetic wastewater simulating the composition of mine drainage No. 56, the removal of Mn and Zn in lab-scale CWs filled with limestone and planted with cattail and common reed has been studied [25]. Under aerobic conditions, Mn is removed from CWs by the biochemical oxidation of soluble Mn to insoluble Mn oxides [26,27]. Mn-oxidizing bacteria and fungi can efficiently oxidize dissolved Mn(II) to Mn(III, IV) through enzymatic catalysis. Furthermore, microalgae can accelerate Mn(II) oxidation through indirect oxidation by increasing the pH and dissolved-oxygen production during their growth [28]. Furthermore, Mn oxides can oxidize and adsorb soluble Mn ions and other metals, such as Zn, Fe, Pb, Ni, and Co [29].

### 3.3.5. Cluster V

Mine drainages classified as Cluster V displayed As concentrations 10-times higher than that of the effluent standard. The predominant forms of arsenic in water are inorganic arsenite ($H_3AsO_3$, As(III)) and arsenate ($HAsO_4^{2-}$, As(V)), with the former being more toxic and less adsorptive. Arsenate is strongly adsorbed on the surface of several common minerals, such as alumina and ferrihydrite. Therefore, the oxidation of As(III) to As(V) is required for effective As removal in mine drainage. The As-removal processes in CWs have been reviewed by several researchers [30]. Although sorption, precipitation, and coprecipitation are the principal processes responsible for the removal of arsenic, bacteria can mediate these processes and play a significant role under favorable environmental conditions.

There are a few mine drainages with high As and Fe concentrations in clusters VI and VII. Due to the fact that the biogeochemical cycles of Fe and As are coupled in natural systems, the presence of Fe affects the speciation of As. Iron oxyhydroxides are especially important and effective for adsorbing and/or coprecipitating As in natural and artificial wetlands.

### 3.3.6. Clusters VI and VII

The Fe concentrations of mine drainages classified into Clusters VI and VII were more than five- and fifteen-times higher than that of the effluent standard, respectively. Under reductive conditions, ferrous iron ($Fe^{2+}$) is the dominant form which completely dissolves in water. However, under oxidative conditions, mine drainage turns reddish brown by spontaneous oxidation to ferric iron, $Fe^{3+}$, and subsequent hydrolysis to ferric hydroxide, $Fe(OH)_3$, or oxyhydroxide, $FeOOH$. Oxyhydroxide precipitation is considered the most important Fe-removal mechanism in CWs [31]. Fe(II) oxidation occurs in the absence of bacteria at pH six or above. Iron-oxidizing bacteria play an important role in Fe oxidation at pH < 4.5. Due to the fact that Fe oxyhydroxides can adsorb/coprecipitate As, iron-oxidizing bacteria may cause the removal of both Fe and As. The co-precipitation of heavy metals with secondary minerals, including hydrous oxides of Fe and Mn, is an important adsorptive mechanism in wetland sediments.

### 3.3.7. Cluster VIII

Mine drainages classified as Cluster VIII demonstrate low pH values of 3–4, with high concentrations of Zn and Pb. Mine drainage No. 59 was treated using lab-scale CWs [32]. The mine drainage contained 12.3 mg/L, 1.3 mg/L, 5.4 mg/L, and 0.15 mg/L of Zn, Pb, Cu, and Cd, respectively, at pH 4.1. In lab-scale CWs filled with limestone or charcoal and planted with common reed, the pH could be raised to sevem or more, and Cu and Pb could be removed to the effluent standard values with a residence time of 24 h. Although the effluent standards were unsatisfactory, 30–50% of Zn and Cd could be mostly removed by soil adsorption. SRB were detected in CWs planted with common reed, suggesting the formation of insoluble metal sulfides, such as ZnS and PbS.

### *3.4. Implications on Design Parameters and Operational Conditions of CWs*

The guidance of METI and JOGMEC proposed four types of CWs: (1) aerobic CWs for mine drainage containing <20 mg/L Fe with pH of >5 under a hydraulic retention time (HRT) of 10–50 h, (2) anaerobic CWs for mine drainage containing Cu, Pb, Zn, and Cd with a pH of 5–8 under a HRT of >15 h, (3) aerobic limestone-based CWs for AMD containing <5 mg/L Fe under a HRT of 2–10 h, and (4) anaerobic limestone-based CWs just for pH neutralization of AMD under a HRT of 2–10 h [5,6]. However, mine-drainage types applicable for the CWs have not been discussed in detail based on the JOGMEC's classifications in the guidance. For establishing design guidelines of CWs, the flow rate and geographical and weather conditions are also important in addition to the mine-drainage classification based on the water quality, as discussed in this study. The design parameters and operational conditions of CWs such as substrate types, plant species, water depth, hydraulic load, HRT, and feeding mode for the mine-drainage treatment should be discussed though a series of laboratory and field experiments.

## 4. Conclusions

Mine drainage in Japan was quantitatively classified into eight types based on water quality: (I) neutral (weakly acidic–weakly alkaline) and low metal concentrations, (II) weakly acidic with low metal concentrations, (III) weakly acidic with a high Zn concentration, (IV) weakly acidic with high Mn and Zn concentrations, (V) acidic with a high As concentration, (VI) acidic with a high Fe concentration, (VII) acidic with an extremely high Fe concentration, and (VIII) acidic with a high Zn concentration. A few exceptional

mine drainages were acidic, Fe- and base-metal-predominant, strongly acidic, and Fe- and As-predominant.

Mine drainages in Cluster VI and partly in Cluster VIII correspond to the conventional JOGMEC type A. A few mine drainages in Clusters V and VII correspond to JOGMEC type B. Cluster IV corresponds to JOGMEC type C. Mine drainages in Cluster V correspond to JOGMEC type D. A few mine drainages of Cluster I correspond to JOGMEC type E. Mine drainages in Clusters II and III did not correspond to any JOGMEC types.

The installation of a CW treatment should be considered with priority to mine drainages with low metal concentrations in Clusters I and II. CWs have been studied just in lab-scales for treating mine drainages in Clusters III, IV, and VIII. The classification of mine drainages implemented in the present study will be helpful in the systematic design and operation of CWs in Japan.

**Author Contributions:** Conceptualization, S.S.; methodology, S.S.; validation, S.S. and T.T.N.; visualization, S.S.; writing—original draft, review and editing, S.S. and T.T.N. All authors have read and agreed to the published version of the manuscript.

**Funding:** This research was funded by JSPS KAKENHI (19K22935).

**Data Availability Statement:** The data used in this study are available from the first author upon reasonable request.

**Acknowledgments:** We are grateful to the Ministry of Economy, Trade, and Industry, Japan for providing the original data.

**Conflicts of Interest:** The authors declare no conflict of interest.

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
