# Peer review of "Classification of Mine Drainages in Japan Based on Water Quality: Consideration for Constructed Wetland Treatments"

_water, doi:10.3390/w15071258_

Round 1
Reviewer 1 Report
Searching for low-cost methods of mine drainage treatment or pre-treatment is one of the most common research directions. The presented classification of 100 mine drainage and assessing the CW construction (model of exploitation) is valuable and can develop the design and exploitation guidelines. In my opinion, the title of the paper should be changed, according to the presented results of the research. There is no information about the application of CW constructions, and the presented results only show the classification of mine drainage according to the water quality. It isn't clear if the presented methods of exploitation CW are the results of the presented research or other experiences. The authors should present more information about water quality. In the text, there are no guidelines for CW construction depending on the type of heavy metals present and their concentrations. Detailed comments: Figure 1. No subtitle Figure 3. IllegibleAuthor Response
We appreciate your kind reviewing for our paper, favorable comments, and helpful suggestions. We carefully revised the paper according to your suggestions.
>In my opinion, the title of the paper should be changed, according to the presented results of the research. There is no information about the application of CW constructions, and the presented results only show the classification of mine drainage according to the water quality. It isn't clear if the presented methods of exploitation CW are the results of the presented research or other experiences. The authors should present more information about water quality. In the text, there are no guidelines for CW construction depending on the type of heavy metals present and their concentrations.
The title was modified to “Classification of Mine Drainages in Japan Based on Water Quality: Consideration for Constructed Wetland Treatments”. The main topic is Classification of Mine Drainages. Consideration for CWs is just a subtopic, although my first interesting was CWs.
Ministry of Economy, Trade, and Industry, Japan has just published a guidance (not guideline) of passive treatment for mine drainage treatment in 2021 [ref. 8, 9]. This paper will be helpful for the future CW guideline in Japan. These were added on L41-42, and L253-266 .
>Detailed comments:
>Figure 1. No subtitle.
It was my careless mistake. The correct title was that “Conceptional mechanisms for metal removal in constructed wetlands.”(p.2, Fig.1)
>Figure 3. Illegible.
The figure was redrawn for legibility. Numbers were drawn bigger I the figure. (p.5, Fig 3)

Reviewer 2 Report
This is the review manuscript entitled “Classification of Mine Drainages in Japan for Considering Application of Constructed Wetland Treatments Based on Water Quality”. I read the paper carefully. As a result, I firmly believe that the paper deserves to be published in Minerals after addressing some major comments, which have been addressed as follows:
1- As you know, AMD is one of the most significant environmental concerns from mining activities worldwide. So, several research works are being conducted on the acid mine drainage topic, while the introduction has been written superficially. Therefore, you must discuss and cite the latest papers in this section. Also, please explain much more about AMD in the first paragraph.
2- Although the number of case studies is high, at least the authors must specify the locations of the cases on Japan’s geographical map.
3- I could not find anything about the methods applied to the dataset. For instance, which methods had been used for determining metal concentrations? What about mineralogical experiments?
4- The authors must present some pictures of acid mine drainage of the cases.
5- Describe much more detail about weather conditions.
6- Sulphate concentration is one of the most vital parameters in generating AMD. So why did you not consider it in your research?
7- A real discussion should be added to the paper.
Author Response
We appreciate your kind reviewing for our paper, favorable comments, and helpful suggestions. We carefully revised the paper according to your suggestions.
1- As you know, AMD is one of the most significant environmental concerns from mining activities worldwide. So, several research works are being conducted on the acid mine drainage topic, while the introduction has been written superficially. Therefore, you must discuss and cite the latest papers in this section. Also, please explain much more about AMD in the first paragraph.
In the first paragraph, it was added that “All major global mining countries have serious problems caused by acid mine drainage (AMD) with a pH value of less than 6.5 [1, 2]” (L.29-30). A reviewing paper for AMD, which was published in 2023, was added as Reference 2 (L299-301).
2- Although the number of case studies is high, at least the authors must specify the locations of the cases on Japan’s geographical map.
By instruction of METI (Ministry of Economy, Trade, and Industry, Japan), names and locations of the mines corresponding to individual drainages are concealed in the present study. Instead, locations of 80 mines in Japan are mapped in reference [2, 3]. This explanation was added on L 84. Thank you for understanding our limitation for data utilization.
3- I could not find anything about the methods applied to the dataset. For instance, which methods had been used for determining metal concentrations? What about mineralogical experiments?
The water quality data set used in this study was measured and provided by METI, Japan (L82-83). Our methodology in this study was not mineralogical experiments but multivariate analysis (L93-95). Thank you for understanding our methodology.
4- The authors must present some pictures of acid mine drainage of the cases.
Regrettably, we do not have enough pictures of mine drainage and cannot show photos that give clues for the location. The data set of METI, Japan also did not include such pictures. Thank you for understanding our limitation for data utilization.
5- Describe much more detail about weather conditions.
Hokkaido and Tohoku belong to the temperate zone with cool humid continental climate. The other regions belong to the temperate zone with humid subtropical climate. Many mines are located in mountains where the climate and weathers are influenced by altitude. These explanations were added on L61, L87-89. The dataset used in this study did not contain any weather information on each mine. But we discussed conditions for the CW design including weather on L260-262.
6- Sulphate concentration is one of the most vital parameters in generating AMD. So why did you not consider it in your research?
I agree that sulfate concentration is important.The data set, provided by METI, included pH values but regrettably did not the sulfate concentration of mine drainage (L110-111).
7- A real discussion should be added to the paper.
Our methodology in this paper was data analysis using an existing dataset provided by METI, without new laboratory experiments. Therefore, the “Results and Discussion”style, was suitable to this paper, rather tahn“Results” and “Discussion”style. The other reviewer allow the style of this paper. Design parameters and operational conditions of CWs cannot be discussed enough only by our data analysis. But we added a discussion section “3.4. Implications on design parameters and operational conditions of CWs” on L253-267.

Round 2
Reviewer 1 Report
Every think is clear.
Reviewer 2 Report
This the re-review manuscript no. 2291841. I checked the revision carefully, and I found that all comments have been addressed appropriately. Therefore, I am pleased to inform you that this version of the article deserves to be published in the Water.